# Statistical Simulation, a Tool for the Process Optimization of Oily Wastewater by Crossflow Ultrafiltration

**DOI:** 10.3390/membranes12070676

**Published:** 2022-06-30

**Authors:** Hajer Aloulou, Afef Attia, Wala Aloulou, Sudip Chakraborty, Lassaad Baklouti, Lasaad Dammak, Raja Ben Amar

**Affiliations:** 1Research Unit “Advanced Technologies for Environment and Smart Cities”, Faculty of Science of Sfax, University of Sfax, Sfax 3038, Tunisia; hajer.aloulou89@yahoo.fr (H.A.); attiaaf@gmail.com (A.A.); walaaloulou6@gmail.com (W.A.); 2Department of DIMES, University of Calabria, Via P. Bucci, Cubo 42/a, 87036 Rende, Italy; sudip.chakraborty@unical.it; 3Department of Chemistry, College of Sciences and Arts at Ar Rass, Qassim University, Ar Rass 51921, Saudi Arabia; blkoty@qu.edu.sa; 4Université Paris-Est Créteil, CNRS, ICMPE, UMR 7182, 2 rue Henri Dunant, 94320 Thiais, France

**Keywords:** ultrafiltration, oily wastewater, heavy metals, response surface methodology, fouling mechanism

## Abstract

This work aims to determine the optimized ultrafiltration conditions for industrial wastewater treatment loaded with oil and heavy metals generated from an electroplating industry for water reuse in the industrial process. A ceramic multitubular membrane was used for the almost total retention of oil and turbidity, and the high removal of heavy metals such as Pb, Zn, and Cu (>95%) was also applied. The interactive effects of the initial oil concentration (19–117 g/L), feed temperature (20–60 °C), and applied transmembrane pressure (2–5 bar) on the chemical oxygen demand removal (RCOD) and permeate flux (Jw) were investigated. A Box–Behnken experimental design (BBD) for response surface methodology (RSM) was used for the statistical analysis, modelling, and optimization of operating conditions. The analysis of variance (ANOVA) results showed that the COD removal and permeate flux were significant since they showed good correlation coefficients of 0.985 and 0.901, respectively. Mathematical modelling revealed that the best conditions were an initial oil concentration of 117 g/L and a feed temperature of 60 °C, under a transmembrane pressure of 3.5 bar. In addition, the effect of the concentration under the optimized conditions was studied. It was found that the maximum volume concentrating factor (*VCF*) value was equal to five and that the pollutant retention was independent of the *VCF*. The fouling mechanism was estimated by applying Hermia’s model. The results indicated that the membrane fouling given by the decline in the permeate flux over time could be described by the cake filtration model. Finally, the efficiency of the membrane regeneration was proved by determining the water permeability after the chemical cleaning process.

## 1. Introduction

Oily wastewater produced from the electroplating industry, consisting of organic materials mixture and heavy metals, is a strong global pollutant that affects the environment and human health [1,2,3,4]. Therefore, it needs to be treated before being discharged into the receiving environment or reused [5]. Removing oil and heavy metals is necessary because they are toxic substances and can cause extensive pollution to water and soil and inhibit the growth of plants and animals. Their effects on human beings are also very dangerous due to the carcinogenic and mutagenic risks that they can produce [6,7].

Oil can be present in wastewater in three forms (droplet size) including free-floating oil (more than 150 μm), unstable dispersed oil (between 20 and 150 μm) and stable emulsified oil (less than 20 μm) [8]. The oil-in-water emulsion is relatively stable due to the presence of surfactants [7]. Surfactants are commonly found in water produced by the oil/gas recovery and metal finishing industries [9]. The permissible limits of oil discharge are ~10 mg/L for inland surface water and ~20 mg/L for marine coastal areas [10]. 

This study is focused on the treatment of real-life oily wastewater contaminated with heavy metal ions, generated by an electroplating industry located in Sfax city (Tunisia). The metallic ions discharged from industries remain in water for a long time as they are not biodegradable [11,12]. The most common toxic heavy metals that are of concern in the treatment of industrial wastewater are zinc (Zn), copper (Cu), mercury (Hg), nickel (Ni), cadmium (Cd), lead (Pb), and chromium (Cr) [13,14,15,16,17,18]. 

Heavy metal ions can naturally be present in the environment, but nowadays, their concentration is high due to increased industrial waste [19]. These toxic ions penetrate the food chain and the human body [20]. Their accumulation in human organs to more than the standard limits can cause serious health-related diseases [21]. Consequently, industries are facing challenges in treating their discharges as the Department of Environment (DOE) imposes limitations on discharges of heavy metals in wastewater via regulations in compliance with the Tunisian standards of wastewater discharge to public sewers (NT106-02) [22].

The nature of wastewater is a critical consideration in the proposed suitable treatment methods for oily and heavy metal removal [8,17]. Conventional methods for separating oily wastewater, such as centrifugation [23], coagulation [24], adsorption [25], electrocatalytic oxidation [26], the Fenton process [27], etc., can be used for treating free-floating oils and dispersed oils. However, most of them are not suitable for treating emulsion with microscopic oil droplets smaller than 20 μm due to the high cost or low treatment efficiency [28]. In this regard, membrane separation technologies using porous ceramic membranes appear to be a highly promising and efficient method for treating oily wastewater containing emulsified oils because of their higher separation efficiency, excellent mechanical resistance, better chemical and thermal stability in harsh environments, ease of processing, long durability, and low maintenance costs [29,30,31,32,33,34]. Researchers have explored numerous treatment technologies to eliminate heavy metals from industrial wastewater. These treatment approaches include adsorption [35,36], membrane filtration [37,38], coagulation–flocculation [39], ion exchange [40], and electrochemical treatment technologies [41,42,43]. Membrane technology has been widely applied to remove heavy metal ions from contaminated water [44] thanks to its relatively low energy consumption and satisfactory treatment performances, and the possibility of recycling with low co-product generation—making this process more efficient and robust [45].

RSM is a numerical approach for multifactorial experimental design analysis and process optimization. This methodology offers a better understanding of the process than standard experimental methods as it can calculate how inputs affect outputs in a complex process that involves interactions between factors [46]. RSM is performed in three steps: the first one requires the analysis of individual and combined parameters. The influence of the primary variables is studied to determine the process’ effectiveness as the second step. The third step involves process optimization using the RSM-based regression model to determine the optimized conditions [47]. In particular, RSM based on BBD is generally utilized thanks to its numerous advantages, such as a lower number of experiments required compared to a three-level full factorial design. Besides this, it is also more successful than central composite design (CCD) [48].

The main objective of this study was to optimize the ultrafiltration process for the elimination of simultaneous oil and heavy metals from electroplating industry wastewater using surface response methodology (RSM) based on a Box–Behnken design (BBD). 

Variations in initial oil concentration (C_oil_), feed temperature values (T), and transmembrane pressure (ΔP) were investigated.COD and stabilized permeate flux were determined to obtain the optimal separation conditions.Statistical analysis of the data was carried out to obtain a suitable mathematical model of the process.Finally, it was found that the model fitted well with the experimental results. The influence of the different factors on the COD retention and the permeate flux was discussed.

## 2. Materials and Methods

### 2.1. Oily Wastewater Collection

Oily wastewater contaminated with heavy metals was collected from an oil separator installed in an electroplating business in Sfax, Tunisia. The characteristics of three different effluents collected over three months are summarized in Table 1. At first, wastewater was pre-filtered using a porous filter paper of 60 μm to remove free-floating oil and solid particles that could clog the membranes.

### 2.2. Ultrafiltration Process

The crossflow ultrafiltration experiments were performed using a semi pilot scale (Figure 1). The installation was equipped with automated systems to control the feed flow rate and temperature. The membrane module contained a tubular UF ceramic multi-channel (7 channel) membrane made from titania purchased from NovaSep, (Miribel, France) with a surface area of 0.155 m^2^ and a 150 kDa separation cut-off. The membrane water permeability was 230 L/h·m^2^·bar. All tests were performed under a transmembrane pressure and temperature ranges from 2 to 5 bar and 20 to 60 °C. The permeate flux was calculated according to the following equation [49]:(1)$$J_{w}=\frac{V}{S\cdot t}$$
where *J_w_* is the permeate flux (L/m^2^ h), *V* is the volume of permeate (L), *S* is the membrane surface area (m^2^), and *t* is the duration of ultrafiltration (h).

The membrane regeneration was accomplished by rinsing the membrane with distilled water and then using an acid–base treatment with an alternative circulation of 2% solutions of NaOH at 80 °C and HNO_3_ at 60 °C for 30 min. Finally, the membrane was washed with distilled water until a neutral pH was obtained. The efficacy of the cleaning protocol was checked by measuring the initial water permeability after the cleaning cycle.

### 2.3. Analytical Methods 

Conductivity and pH were measured by a conductivity meter (EC-400L, Istek, Seoul, Korea) and a pH meter (pH-220L, Istek). Turbidity was measured by a turbidity meter (model 2100A, Hach) agreeing with standard method 2130B. The COD was determined by a colorimetric technique (COD 10119, Fisher Bioblock Scientific, Illkirch, France). The oil and heavy metal retention content was measured by determining the feed and solution concentrations using a UV-spectrophotometer (UV-9200, Beijing, China) at a wavelength of 363 nm and atomic absorption spectroscopy (AAS, PerkinElmer, Waltham, MA, USA), respectively.

For the evaluation of UF rejection, the rejection of different parameters (COD, turbidity, oil, and heavy metals) was determined by Equation (2) [50,51]:(2)$$R(\%)=\frac{\left(C_{f}-C_{p}\right)}{C_{f}}\times 100$$
where *C_f_* and *C_p_* represent the concentration of pollutants in the feed and in the permeate, respectively.

### 2.4. Experimental Design Methodology 

The response surface methodology model (RSM) was applied to evaluate the effects of ultrafiltration parameters and to optimize various conditions for different responses. Table 2 summarizes the studied variables: initial oil concentration (X_1_), temperature (X_2_), and transmembrane pressure (X_3_).

A Box–Behnken experimental design (BBD) with three numeric factors over three levels was studied [51]. The BBD included 13 randomized runs with one replicate at the central point. The matrix, experimental range, and responses are presented in Table 3.

RSM is a statistical method for the multifactorial analysis of experimental data that supplies a higher understanding of the process than standard methods of experimentation due to its ability to predict how inputs affect outputs in a complex process where different factors can interact among themselves. All the other polynomial equation coefficients were tested for significance with an analysis of variance (ANOVA) [52]. For responses obtained after the experiments (R COD and permeate flux), a polynomial model of the second degree was established to evaluate and quantify the influence of the variables as follows:(3)$$Y(\%)=b_{0}+\sum b_{i}X_{i}+\sum \sum b_{ij}X_{i}X_{j}+\varepsilon ;\;\;\;\;\;i\neq j$$
where *X_i_* and *X_j_* are the coded variables (−1 or +1), *b*_0_ is the mean of the responses obtained, *b_i_* is the main effect of factor *i* for the response *Y*, *b_ij_* is the interaction effect between factors *i*, and *j* represents the error in the response.
(4)$$Y(\%)=b_{0}+\sum b_{i}X_{i}+\sum \sum b_{ij}X_{i}X_{j}+\sum \sum b_{ii}X_{i}{}^{2}+\varepsilon ;\;\;\;\;\;i\neq j$$
where *Y*, *b*_0_, *b_i_*, *b_ii_*, *b_ij_*, *X_i_*, and *X_j_* represent the predicted response, the constant coefficient, the linear coefficient, the interaction coefficient, the quadratic coefficient, and the coded values of the factors, respectively.

The sufficiency of the model was determined by the coefficient of determination (R^2^) and *p*-value. The statistical analysis was evaluated using Design-Expert 12 software. Response surface plots were indicated for two factors, where the third factor was set to its medium value.

### 2.5. Investigation of the Fouling Mechanism

To determine the fouling mechanism that occurred during the UF of the oily wastewaters, a mathematical model established by Hermia [53] was applied. This model is based on conventional constant pressure dead-end filtration equations; it has been widely evaluated in crossflow filtration studies [54] and has been used to predict decreases in flux during the MF and UF of oil-in-water emulsions [55,56,57,58]. The equation of the model is expressed by Equation (5) [53] as follows: (5)$$\frac{d^{2}t}{dV^{2}}=K{\left(\frac{dt}{dV}\right)}^{n}$$
where *V* is the permeation volume, *t* is the filtration time, *K* is a constant, and *n* is a value illustrating the different fouling mechanisms (Table 4). The Hermia model is based on four empirical approaches: complete pore blocking, standard pore blocking, intermediate pore blocking, and cake filtration.

In a complete blocking model, each pollutant particle blocks a pore of the membrane without overlapping on top of any other. In the standard blocking model, the size of the particle is smaller than the pore diameter; consequently, the foulant particles can enter the pores and form a deposit on the pore walls, which reduces the pore volume. In the intermediate blocking model, some pollutant particles are in direct contact with the pores, but a number of them are on top of others. In the cake filtration model, many foulant particles accumulate on the membrane surface and create a cake layer, forming an additional resistance to the permeate flux [7].

The correlation of the experimental permeate flux decline data with the above fouling mechanisms was studied by comparing the correlation coefficient R^2^ values reported from the linear regression analysis using Equations (6)–(9) (Table 4). A higher R^2^ correlation coefficient equation corresponds to the dominant membrane fouling mechanism.

## 3. Results and Discussion

### 3.1. UF Experiments

The efficiency of the UF of the industrial oily wastewater contaminated with heavy metals using a ceramic membrane (150 KDa) was not determined only on the basis of the observed stabilized permeate flux but also concerning the retention of different parameters (oil, turbidity, COD, and heavy metals). It is worth noting that an almost total retention of oil and turbidity and a high elimination of heavy metals such as Pb, Zn, and Cu (>95%) were achieved by the UF process regardless of the initial pollutant values and the treatment conditions. The COD removal and permeate flux results show that they were affected by different parameters such as the initial oil concentration, the feed temperature, and the applied transmembrane pressure.

### 3.2. COD Removal Response

Table 5 illustrates the regression coefficients obtained by the ANOVA of a quadratic model for COD removal and the modified quadratic model for permeate flux. The *p*-value determined the significance of the input factors and their interactions in the studied model. A factor affects the response if the *p*-value is less than the used probability level. The significance was judged at probability levels less than 0.05 [59].

Table 5 shows the mathematical model that explains the relationship between responses and dependent and independent variables represented by oil concentration (X_1_), temperature (X_2_), transmembrane pressure (X_3_), and the significance level of the linear and quadratic models.

In line with Joglekar et al. [60], who proved that the model fit is good when R^2^ > 0.80, the R^2^ value coefficient of 0.985 confirmed the agreement of the mathematical model with the experimental data and showed that the model fit was significant.

Furthermore, R^2^ evaluates the discrepancy or variance in the apparent values, which could be explained by the independent variables and their interactions rather than the design of specific factors. In fact, R^2^ = 0.985 shows that the model could describe 98.5% of the total response variation and that only 1.5% of it cannot be explained by the empirical model. As a result, the model equation was better at representing the COD removal regarding the three independent variables. The comparison of the experimental results (actual values) and the predicted values by the model is presented in Figure 2. The theoretical and empirical values were very close for the COD removal. This proximity reflects the robustness of the statistical models used. 

In Figure 3, the experimental results prove that the removal of COD was strongly affected by the three independent variables represented by initial oil concentration, temperature, and transmembrane pressure. Furthermore, almost total oil retention was observed whatever the conditions of the UF treatment were.

### 3.3. Permeate Flux Response

The effects of the input factors on permeate flux values were given and analyzed. The modified quadratic model proved that the linear model terms of initial oil concentration (X_1_) and temperature (X_2_), as well as the quadratic model of the term X_12_, were significant (*p*-value < 0.05). The optimized model showed that the permeate flux was only affected by the initial oil concentration and the temperature as the applied transmembrane pressure did not affect the permeate flux. This estimated result correlated with the experimental results, showing that the permeate flux was almost stable, at around 103 L/h·m^2^ for a pressure of 3 bar under the experimental conditions: C_oil_ = 68 g/L, and T = 20 °C (Figure 4). 

The relatively high R^2^ (0.901) value confirms that the model fit the data well. Additionally, this coefficient measures the variability in the observed response values, which can be described by the independent factors and their interactions over the range of the corresponding factors; it indicated that the model could describe 90.1% of the total variation—only 9.9% of it was not described. Figure 5 suggests that the experimental results for the permeate flux value were not close enough to the predicted value.

### 3.4. Optimization of COD Removal and Permeate Flux

The optimizations by RSM were performed by maximizing the COD removal and permeate flux. In Figure 6 and Figure 7, the responses can be observed from the three-dimensional surfaces obtained with the proposed quadratic degree model. The interactions of independent variables with the treatment of the oily wastewater were investigated. The initial oil concentration (19–117 g/L), the feed temperature (20–60 °C), and the transmembrane pressure (2–5 bar) were evaluated. According to the results illustrated in Table 3 and Figure 6 and Figure 7, it is clear that the maximum COD removal (97%) and the highest permeate flux (232 L/h·m^2^) were obtained at the optimal conditions of C_oil_ = 117 g/L, T = 60 °C, and ΔP = 3.5 bar by applying the RSM model. From Figure 6 and Figure 7, it can be observed that the model is highly desirable, since the predicted values for the COD removal and permeate flux were 96.57% and 226.26 L/h·m^2^, respectively.

Based on Table 6, different methods for the optimization of UF processes such as Box–Behnken experimental design (BBD), central composite design (CCD), central composite rotatable design (CCRD), and the Taguchi method have been applied in many previous works. The optimized responses obtained in this study by the BBD method were close to some other reactions reported in the literature determined by using BBD or CCD methods [61,62]. Our results confirm that the BBD model achieved higher response values in terms of COD removal and permeate flux compared to results reported by the literature using other models [63,64,65,66].

### 3.5. Effect of Concentration

The UF experiments were carried out by recycling the retentate and recovering the permeate at optimized conditions of treatment as follows: C_oil_ = 117 g/L, T = 60 °C, and ΔP = 3.5 bar. Figure 8 represents the variation of the permeate flux as a function of the volume concentrating factor (*VCF*).

In concentration mode (without recirculation of the permeate), the mass balance is determined using the following classical equation:(10)$$V_{i}C_{i}=V_{p}C_{p}+V_{r}C_{r}$$
where: *V_i_*, *V_p_*, and *V_r_* are the initial, permeate, and retentate volumes, respectively; and *C_i_*, *C_p_*, and *C_r_* are the initial oil concentration, oil concentration in the permeate, and oil concentration in the retentate, respectively

On the other hand, the volume balance is given by Equation (11):(11)$$V_{i}=V_{p}+V_{r}$$

Considering that:

-The oil retention was determined by: (12)$$R(\%)=(1-\frac{C_{p}}{C_{i}})\times 100$$

-The concentration factor (*CF*) and the volume concentration factor (*VCF*) are given by: (13)$$CF=\frac{C_{r}}{C_{i}}$$
(14)$$VCF=\frac{V_{i}}{V_{r}}$$

Equations (12)–(14) can be combined to obtain the following equation: (15)$$CF=\frac{1}{\left(1-(1-\frac{1}{VCF})R\right)}$$

For *R* = 100%, as is the case here, total retention of the oil is shown—i.e., *C_p_* = 0; consequently, *CF* = *VCF*.

The maximum *VCF* value observed in this case was equal to five. Indeed, the permeate flux decreased slightly from 232 L/h·m² at *VCF* = 1 to 212 L/h·m² at *VCF* = 5, then it decreased quickly to 171 L/h·m² at a *VCF* of 6.2. A negligible flux reduction was present of around 8.6% between a *VCF* of 1 and a *VCF* of 5. At a *VCF* of 6, the decrease in the flux was significantly (up to 26%) associated with membrane fouling—mainly due to the concentration of pollutants near the membrane surface [67]. Figure 9 shows a high retention of contaminants in terms of COD, oil, and heavy metals of up to 94%, whatever the FCV value range (from 1 to 6).

### 3.6. Application of the Hermia Model

The accumulation of oil and suspension at the membrane surface causes a rapid decrease in the permeate flux. The determination of the flux decline during fouling is critical for ultrafiltration processes. Four filtration models including complete pore blocking, standard pore blocking, intermediate pore blocking, and cake filtration evaluated the flux decline mechanism [65]. Figure 10a–d illustrate the different pore blocking models for UF of the oily industrial wastewater by a ceramic TiO_2_ membrane at optimal treatment conditions, as follows: C_oil_ = 117 g/L, T = 60 °C, and ΔP = 3.5 bar. According to the R^2^ values, it appears that the formation of the cake layer model resulted in slightly higher R^2^ values in comparison to the other fouling mechanisms; therefore, it can be chosen as the best model to describe the fouling mechanism. As a result, it can be expected that the majority of the particles in the feed solutions were bigger compared to the membrane pores. Consequently, accumulated molecules on the membrane surface increased the resistance to the permeate flux [68,69,70].

### 3.7. Cleaning Study

After concentration tests at optimized conditions, the results confirmed intensive membrane fouling (>26%). For this reason, to recover the initial membrane performance, an acid–base cleaning procedure was required [71]. The efficiency of the membrane regeneration was determined by checking the water permeability. Figure 11 presents the evolution of the water permeate flux with the transmembrane pressure for the virgin and the regenerated membranes. The results demonstrated that the water permeability values were very close, confirming the efficiency of the cleaning process used.

## 4. Conclusions

The objective of this study was to determinate the best conditions for the treatment of industrial wastewater contaminated with oil and heavy metals, using the response surface methodology. The obtained results revealed that BBD for the RSM model was effectively useful for this application. The UF process achieved the almost total retention of oil and turbidity and a high removal of heavy metals such as Pb, Zn, and Cu (>95%), independently of the initial values and treatment conditions. However, the COD removal and permeate flux were mainly affected by the initial oil concentration, feed temperature, and applied transmembrane pressure. The optimized conditions were 117 g/L, 60 °C, and 3.5 bar. Under these conditions, 97% COD removal and 232 L/h·m^2^ permeate flux were achieved experimentally, and a maximum volume concentrating factor (*VCF*) of five was obtained. The results also revealed that the different pollutant retention values were independent of the *VCF*. Moreover, Hermia’s model was applied to assess the membrane fouling mechanism. The data was in agreement with the cake layer model. The chemical cleaning process allowed the complete restoration of the initial water membrane permeability.

This study shows that the UF process is an efficient method for the simultaneous elimination of oil and heavy metals from industrial wastewater. Furthermore, the response surface methodology is very useful for modeling and optimizing membrane treatments. 

## Figures and Tables

**Figure 1 membranes-12-00676-f001:**
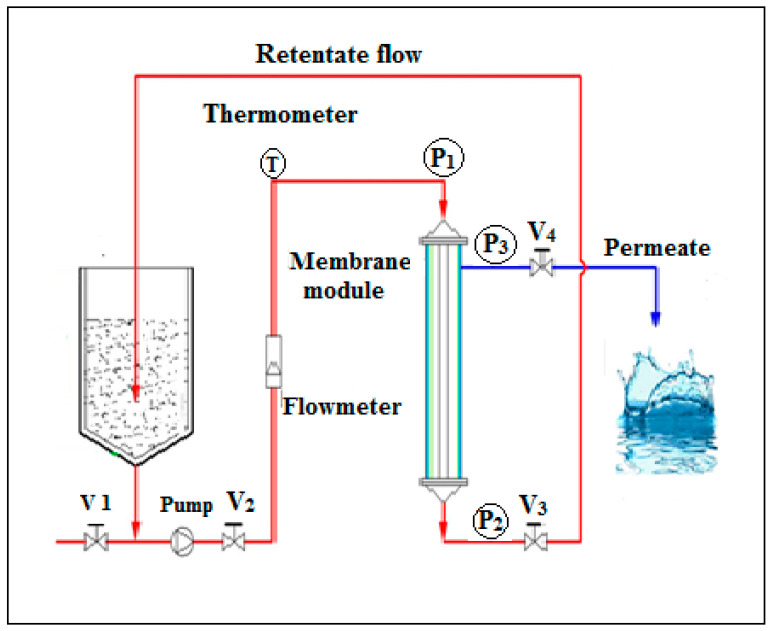
Schematic representation of crossflow Ultrafiltration experiment set-up.

**Figure 2 membranes-12-00676-f002:**
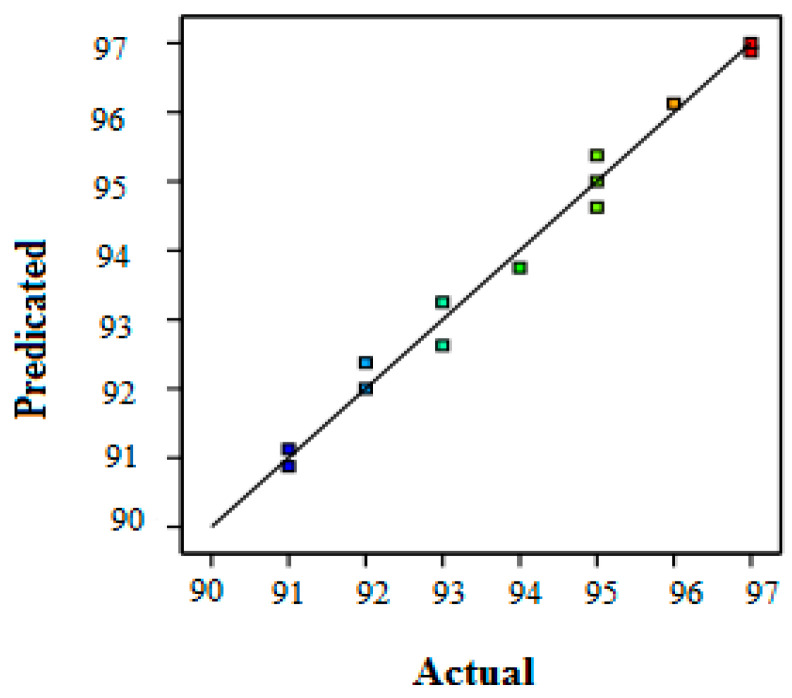
Comparison of calculated and predicted values for COD removal by RSM.

**Figure 3 membranes-12-00676-f003:**
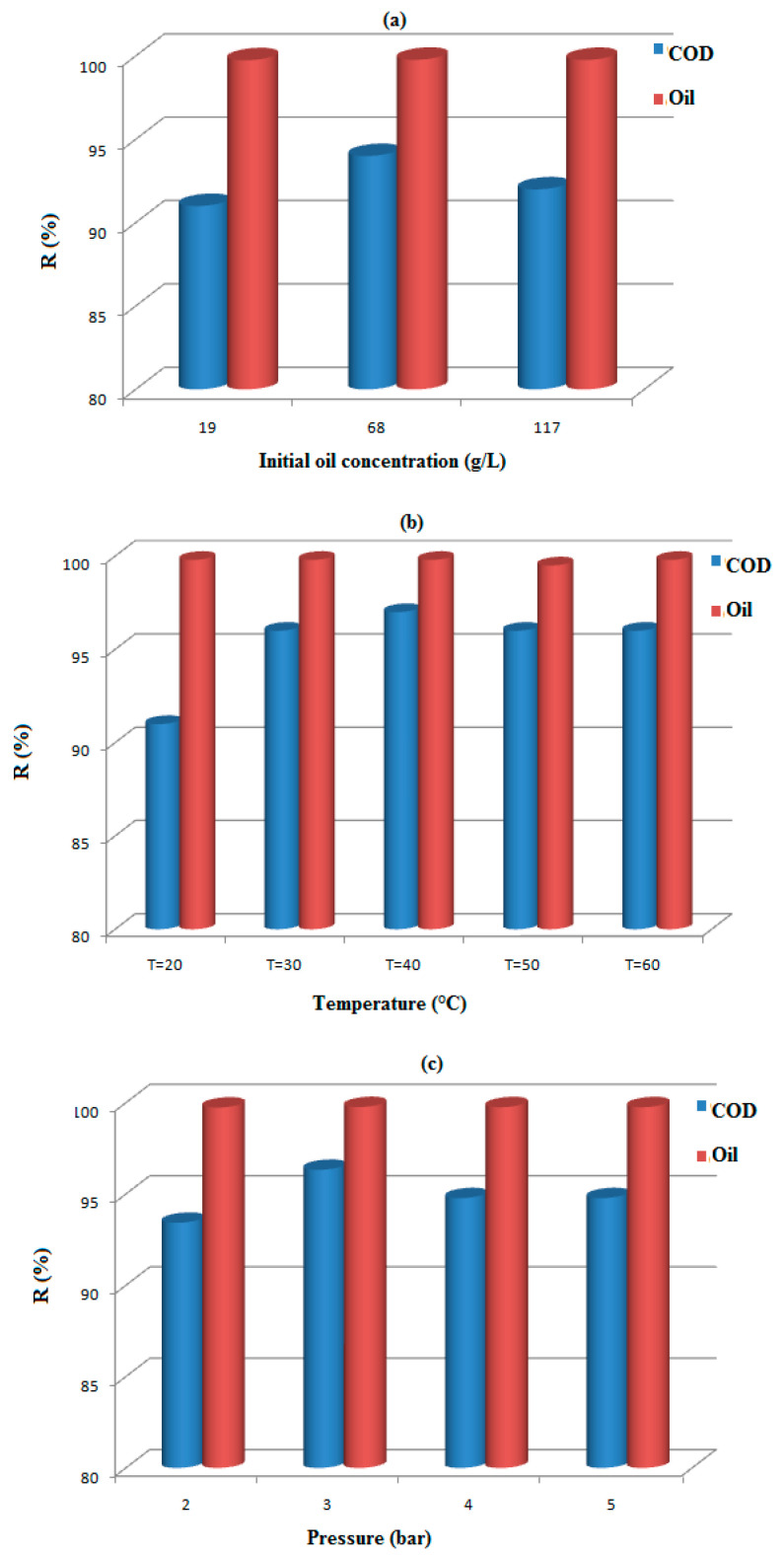
Retention of Oil and COD versus: Initial oil concentration (**a**), Temperature (**b**), Pressure (**c**).

**Figure 4 membranes-12-00676-f004:**
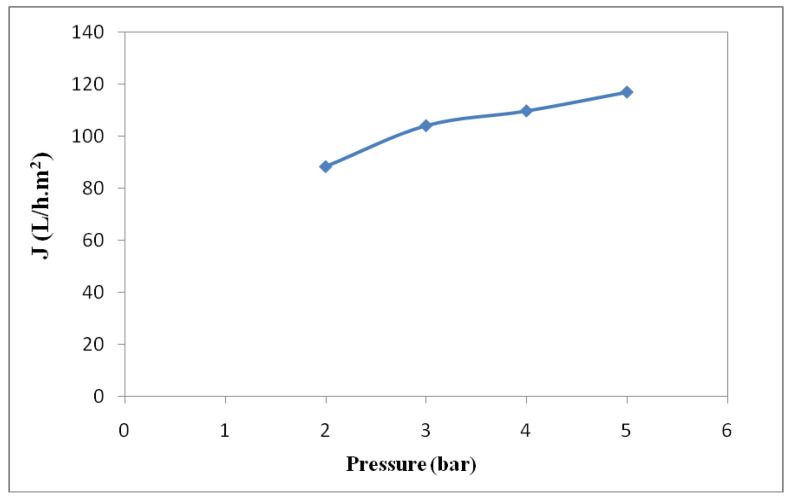
Evolution of stabilized permeate flux with the applied pressure at a C_oil_ of 68 g/L, T = 20 °C.

**Figure 5 membranes-12-00676-f005:**
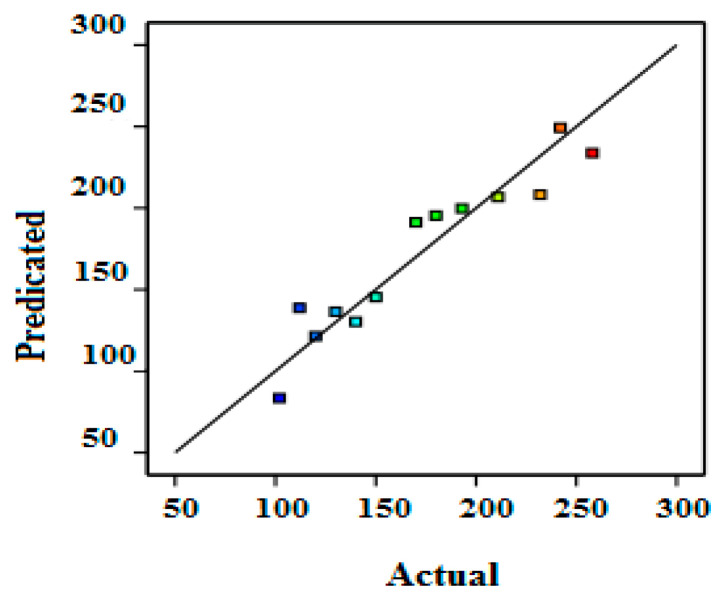
Comparison of the calculated and predicted values for permeate flux by RSM.

**Figure 6 membranes-12-00676-f006:**
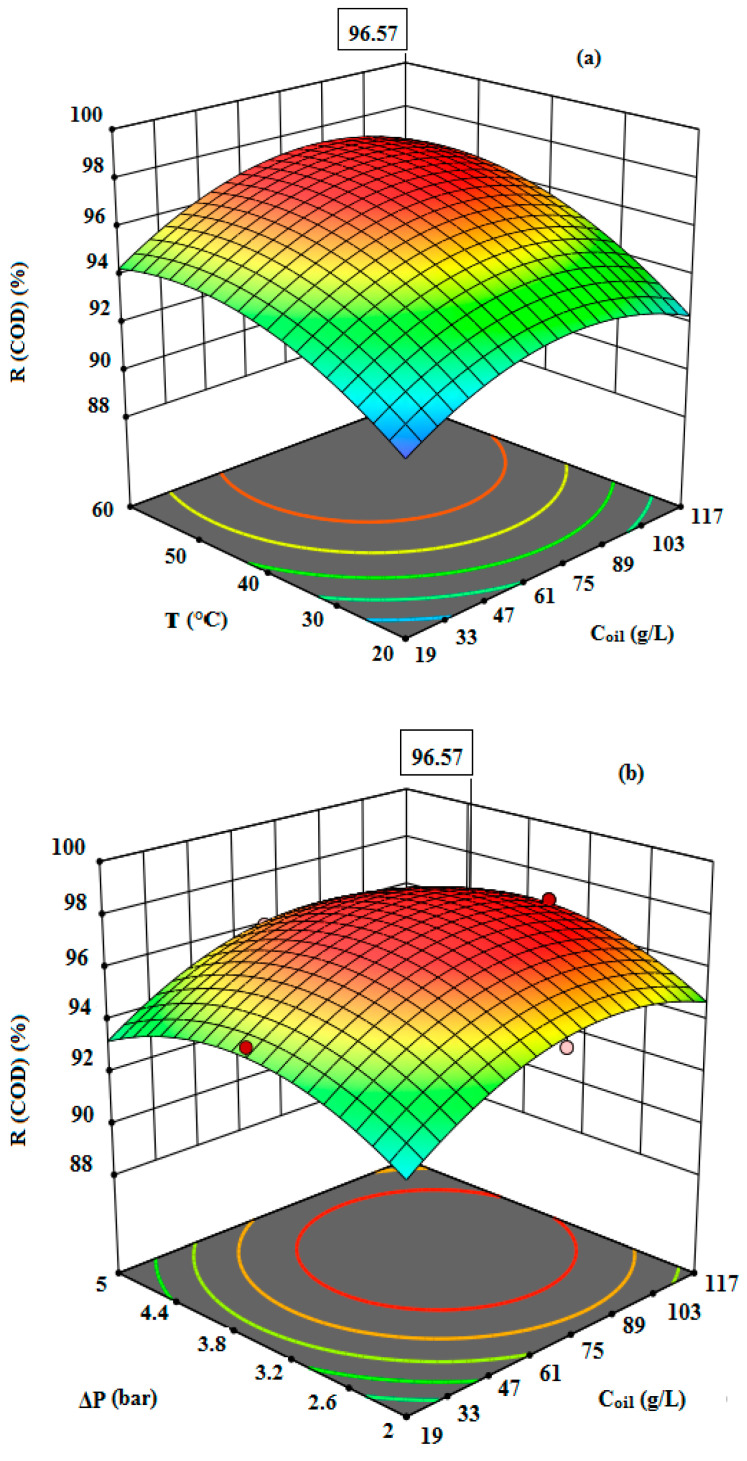
The response surface plots showing the effects of variables on COD removal: The interaction of initial oil concentration and temperature (**a**), the interaction of initial oil concentration and transmembrane pressure (**b**), the interaction of temperature and transmembrane pressure (**c**).

**Figure 7 membranes-12-00676-f007:**
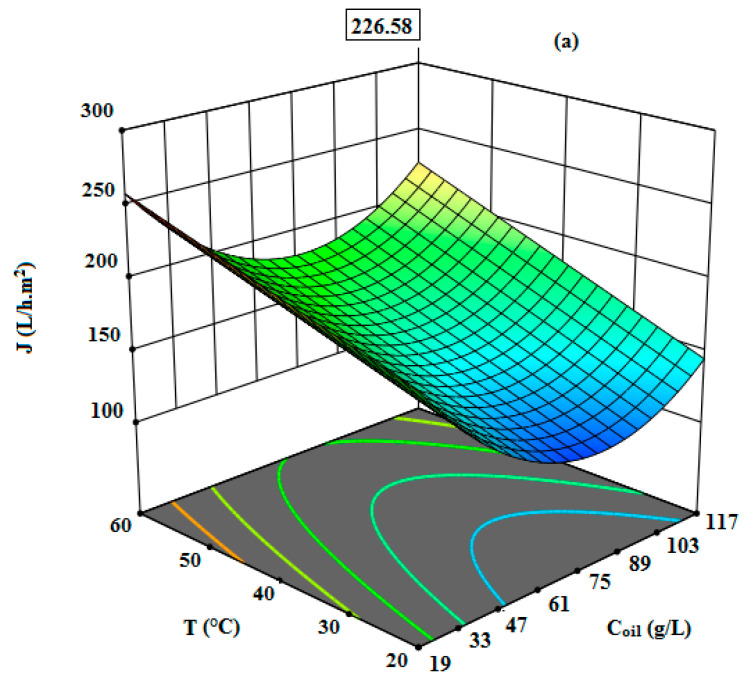
The response surface plots showing the effects of variables on permeate flux: The interaction of initial oil concentration and temperature (**a**), the interaction of initial oil concentration and transmembrane pressure (**b**), the interaction of temperature and transmembrane pressure (**c**).

**Figure 8 membranes-12-00676-f008:**
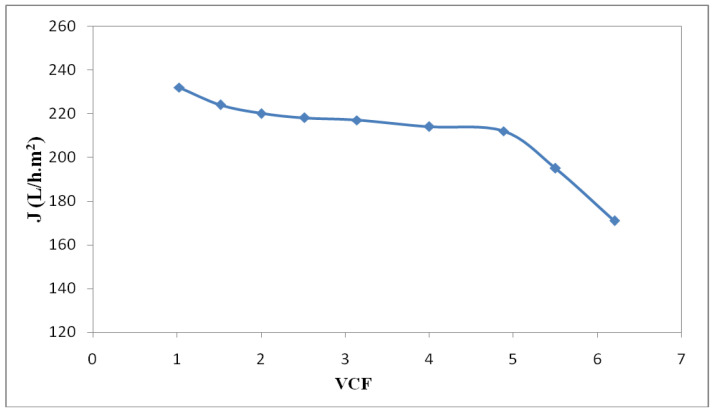
Evolution of permeate flux versus *VCF* at optimized conditions of treatment: C_oil_ of 117 g/L, T = 60 °C, and ΔP = 3.5 bar.

**Figure 9 membranes-12-00676-f009:**
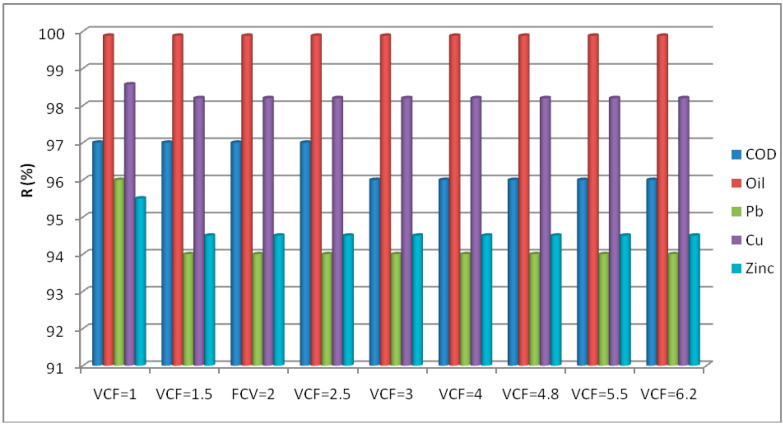
Retention of different pollutants with *VCF*.

**Figure 10 membranes-12-00676-f010:**
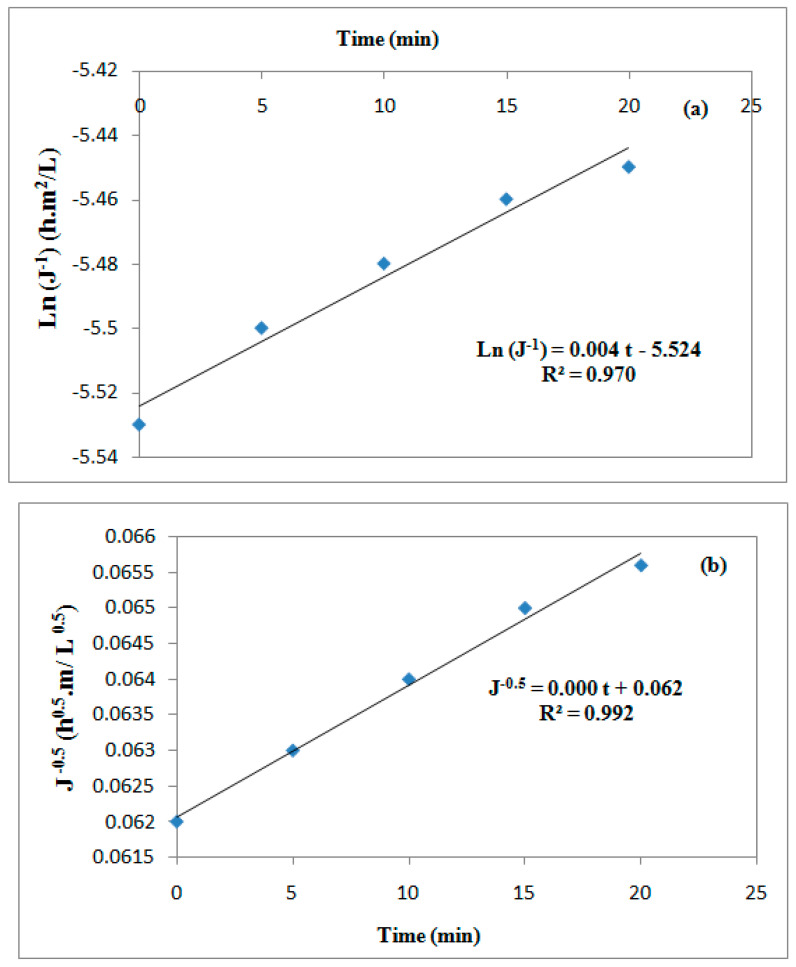
Linearized models of permeate fluxes of wastewater using a UF TiO_2_ membrane: complete pore blocking (**a**), standard pore blocking (**b**), intermediate pore blocking (**c**), and cake filtration (**d**).

**Figure 11 membranes-12-00676-f011:**
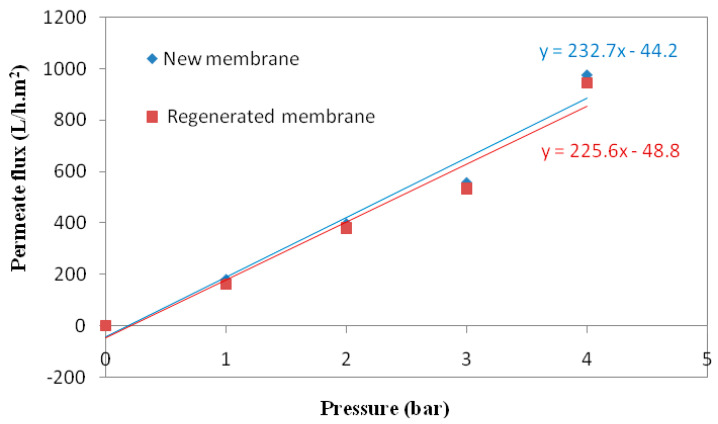
Water permeate flux versus transmembrane pressure for the new and regenerated membrane.

**Table 1 membranes-12-00676-t001:** Physicochemical characteristics of the industrial oily wastewater contaminated with heavy metals collected over three months.

Parameters	Unity	Sample 1	Sample 2	Sample 3
pH	-	7.14 ± 0.2	8.12 ± 0.2	6.9 ± 0.2
Conductivity	mS/cm	4.66 ± 0.4	4.11 ± 0.4	3.33 ± 0.4
Turbidity	NTU	3610 ± 100	>6000 ± 100	>6000 ± 100
COD	mg/L	1125 ± 200	4950 ± 200	8175 ± 200
Oil contents	g/L	19 ± 1	68 ± 1	117 ± 1
Copper	mg/L	2.63 ± 0.02	2.74 ± 0.02	4.1 ± 0.02
Lead	mg/L	16 ± 0.02	43.8 ± 0.02	21.5 ± 0.02
Zinc	mg/L	3.6 ± 0.02	2.5 ± 0.02	10.4 ± 0.02
Nickel	mg/L	<0.1 ± 0.001	<0.1 ± 0.001	<0.1 ± 0.001
Chromium	mg/L	<0.02 ± 0.001	<0.02 ± 0.001	<0.02 ± 0.001

**Table 2 membranes-12-00676-t002:** Variables and factor levels in the Box–Behnken experimental design.

Input Factors	Variables	Factor Levels		
		−1	0	1
C_oil_ (g/L)	X_1_	19	68	117
T (°C)	X_2_	20	40	60
ΔP (bar)	X_3_	2	3.5	7

**Table 3 membranes-12-00676-t003:** Box–Behnken experimental design and responses.

Run	Input Factors			Responses	
	C_oil_ (g/L)	T (°C)	ΔP (bar)	R COD (%)	Permeate Flux (L/ h·m²)
1	68	20	5	93	120
2	19	60	3.5	95	242
3	68	20	2	91	102
4	117	40	2	94	112
5	117	60	3.5	97	232
6	19	40	2	92	211
7	68	40	3.5	97	130
8	68	60	5	96	180
9	117	20	3.5	92	140
10	19	20	3.5	91	170
11	19	40	5	93	258
12	68	60	2	95	150
13	117	40	5	95	193

**Table 4 membranes-12-00676-t004:** Fouling mechanisms based on the Hermia model.

Fouling Mechanism	N	Linearized Form		Schematic Diagram
Complete pore blocking	2	$Ln(J_{w}^{-1})=Ln(J_{0}^{-1})+K_{b}t$	(6)	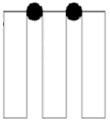
Standard pore blocking	1.5	$J_{w}^{-0.5}=J_{0}^{-0.5}+K_{S}t$	(7)	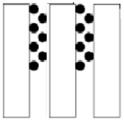
Intermediate pore blocking	1	$J_{w}^{-1}=J_{0}^{-1}+K_{i}t$	(8)	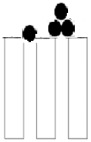
Cake filtration	0	$J_{w}^{-2}=J_{0}^{-2}+K_{c}t$	(9)	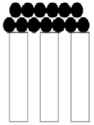

**Table 5 membranes-12-00676-t005:** Estimated coefficients for Permeate flux and R (COD) responses.

	b_0_	b_1_	b_2_	b_3_	b_12_	b_13_	b_23_	b_11_	b_22_	b_33_
R(COD)	97	0.875	2	0.625	0.25	1.6992 10^−17^	−0.25	−1.75	−1.5	−1.75
*p*-values		**0.0158**	**0.0015**	**0.0385**	0.3910	1.0000	0.3910	**0.0132**	**0.0201**	**0.0132**
Permeate Flux	136.4	−25.5	34	22	5	8.5	3	58.35		
*p*-values		**0.0354**	**0.0124**	0.0566	0.7080	0.5300	0.8213	**0.0097**		

**Table 6 membranes-12-00676-t006:** Comparison of the UF membrane, optimization method, optimal factors, and responses.

UF Membrane Material	Experimental Design Method	Optimal Factors	Responses	References
TiO_2_	BBD	-Initial oil concentration: 117 g/L -Temperature: 60 °C -Transmembrane Pressure: 3.5 bar	-COD removal: 97% -Permeate flux: 232 L/h·m^2^	This work
Nanocomposite	CCD	-Transmembrane pressure: 3 bar -pH: 9.0 -Feed concentration: 600 ppm	-Water flux: 152 L/h·m^2^ -Oil rejection: 98.72%	[61]
Mullite	BBD	-pH: 7.2 -Feed concentration: 921 mg/L -Coagulant concentration: 207 mg/L	-Water flux: 123.85 L/h·m^2^ -Oil rejection: 97.31%	[62]
Hollow fiber polyvinylidene fluoride	CCD	-Transmembrane pressure: 1 bar -Velocity 3 m/s	-Permeate flux: 50 L/h·m^2^ -Turbidity removal: 79% -COD removal: 77%	[63]
Anionic polyacrylamide (APAM)	CCRD	-C APAM and C_oil_: <50 mg/L -pH < 4 -Transmembrane pressure: <0.075 MPa	-Minimum relative flux J/J_0_ = 4%	[64]
Al_2_O_3_–ZrO_2_	Taguchi method	-Transmembrane pressure: =5 bar -pH = 7 -Oil concentration: 0.5% *v*/*v*	-Permeate flux: 55.441 L/h·m^2^	[65]
γ-Al_2_O_3_	BBD	-Feed temperature: 35 °C -Transmembrane pressure: 5 bar -Crossflow velocity CFV: 0.735 m/s	-Permeate flux: 112.7 kg/h·m^2^	[66]

## Data Availability

The data presented in this study are available on request from the corresponding author.

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
