# Peer review of "Statistical Simulation, a Tool for the Process Optimization of Oily Wastewater by Crossflow Ultrafiltration"

_membranes, 2022, doi:10.3390/membranes12070676_

Round 1

Reviewer 1 Report

Abstract: replace “wastewater” by “wastewater treatment”

Abstract: I will suggest the oily be replaced by industrial since the later could be very confusing as there are different types of oil. Besides, heavy metals in the wastewater were also included in the investigation.

Line 53: generated from an electroplating industry, located in the Sfax city (Tunisia).

Line 110: More information is required on the sample collection. What did sample 1, sample 2, and sample 3 represent? Does each sample represents collection for each month? Were they collected from the same point at the electroplating industry?

Line 160: Three variables were selected for optimization. Why are only these variables selected? Additional information on the choice of variables selection is required.

Author Response

Comment 1: Abstract

- Replace “wastewater” by “wastewater treatment”

- I will suggest the oily be replaced by industrial since the later could be very confusing as there are different types of oil. Besides, heavy metals in the wastewater were also included in the investigation.

Answer: We appreciate the reviewer comment, and we considered these suggestions in the revised paper.

Comment 2, Line 53: generated from an electroplating industry, located in the Sfax city (Tunisia).

Answer: We appreciate the reviewer suggestion and we added “,” in the revised paper.

Comment 3, Line 110: More information is required on the sample collection. What did sample 1, sample 2, and sample 3 represent? Does each sample represents collection for each month? Were they collected from the same point at the electroplating industry?

Answer: We agree with the reviewer and more information is added in the revised paper.

In fact, sample 1, sample 2, and sample 3 represent 3 samples collected in different 3 dates from an oil separator installed in the electroplating industry.

Comment 4, Line 160: Three variables were selected for optimization. Why are only these variables selected? Additional information on the choice of variables selection is required.

Answer:

We appreciate the reviewer comment. In fact, we chose these 3 variables: initial oil concentration, feed Temperature and applied transmembrane pressure for the optimization because they have an effect on the membrane treatment process independently.

Reviewer 2 Report

The authors need to improve the quality and font size of figures 4, 6, 7, 8, and 10. The font size in these figures is very small. Moreover, good color selection can improve the quality of the figure. 

Author Response

Comment: The authors need to improve the quality and font size of figures 4, 6, 7, 8, and 10. The font size in these figures is very small. Moreover, good color selection can improve the quality of the figure. 

Answer: We agree with reviewer, and we have improved the quality of these figures.

Reviewer 3 Report

Authors have presented an innovative research in this article. Paper is satisfactory for the reader point of view. Few points needed to be addressed.

1] Few changes in abstract is required in 30-35 line number.

2]Line number 94-101, can be presented with bullets points.

3] Justification of eq. 1.

4] Table 2 data is not satisfactory , can be improved.

5] Eq.3, 4 font size is different.

6] what is the requirement and justification of table 4.

7] Line 283-285 needs modification.

8] subsection 2.5 needs more explanation.

9] conclusion is not satisfactory. 

Author Response

Comment 1: Few changes in abstract is required in 30-35 line number.

Answer: We appreciate the reviewer comment and we have improved these lines in the revised paper.

Comment 2: Line number 94-101, can be presented with bullets points.

Answer: We agree with reviewer and this remark was taken into the revised paper.

Comment 3: Justification of eq. 1.

Answer: We appreciate the reviewer comment. The eq.1 represents how the permeate flux is calculated. All details are presented in the text with a reference.

Comment 4: Table 2 data is not satisfactory, can be improved.

Answer: We agree with reviewer and Table 2 was improved.

Comment 5: Eq.3, 4 font size is different.

Answer: We agree with reviewer and this remark was taken into the revised version.

Comment 6: What is the requirement and justification of table 4.

Answer: We appreciate the reviewer comment. Table 4 represents the different equations and their schematic representations following Hermia’s model to better explain the fouling membrane phenomenon.

Comment 7: Line 283-285 needs modification.

Answer: We agree with reviewer. We have taken into account this remark in the revised paper (Line 308-310).

Comment 8: subsection 2.5 needs more explanation.

Answer: We appreciate the reviewer comment. For the authors Herima’s model was clearly explained in this subsection 2.5. Some references were also given to support the use of this model in our case.

Comment 9: conclusion is not satisfactory

Answer: Thanks for this remark. The conclusion was improved.